# Can Land Cover Changes Mitigate Large Floods? A Reflection Based on Partial Least Squares-Path Modeling

**Daniela Patrícia Salgado Terêncio [1], Luís Filipe Sanches Fernandes [1], Rui Manuel Vitor Cortes [1], João Paulo Moura [1] and Fernando António Leal Pacheco [2,\*]**

[1] Centro de Investigação e Tecnologias Agroambientais e Biológicas, Universidade de Trás-os-Montes e Alto Douro, Ap 1013, 5001–801 Vila Real, Portugal. dterencio@utad.pt (D.P.S.F.); lfilipe@utad.pt (L.F.S.F.); rcortes@utad.pt (R.M.V.C.); jpmoura@utad.pt (J.P.M.)

[2] Centro de Química de Vila Real, Universidade de Trás-os-Montes e Alto Douro, Ap 1013, 5001–801 Vila Real, Portugal

\* Correspondence: fpacheco@utad.pt; Tel.: +00-259-350280

**Abstract:** Common approaches to large flood management are Natural Water Retention Measures and detention basins. In this study, a Partial Least Squares-Path Model (PLS-PM) was defined to set up a relationship between dam wall heights and biophysical parameters, in critical flood risk zones of continental Portugal. The purpose was to verify if the heights responded to changes in the biophysical variables, and in those cases to forecast landscape changes capable to reduce the heights towards sustainable values (e.g., <8 m). The biophysical parameters comprised a diversity of watershed characteristics, such as land use and geology, surface runoff, climate indicators and the dam heights. The results have shown that terrain slope (w > 0.5), rainfall (w > 0.4) and sedimentary rocks (w > 0.5) are among the most important variables in the model. Changes in these parameters would trigger visible changes in the dam wall height, but they are not easily or rapidly modified by human activity. On the other hand, the parameters forest occupation and runoff coefficient seem to play a less prominent role in the model (w < 0.1), even though they can be significantly modified by human intervention. Consequently, in a scenario of land cover change where forest occupation is increased by 30% and impermeable surfaces are decreased by 30%, interferences in the dam heights were small. These results open a discussion about the feasibility to mitigate large floods using non-structural measures such as reforestation.

**Keywords:** flood risk attenuation; PLS-SEM; detention basin; mitigation strategies; landscape change

## 1. Introduction

One of the consequences of climate change is the increase of spatial and temporal water variability as well as extreme events, in frequency and intensity [1,2]. Floods are among the most destructive water-related hazards and are the greatest economic natural disaster that occur in Europe via damage property and infrastructure, as well as physical injury and loss of human lives [3]. Nowadays, technical means for controlling extreme floods remain limited, fosters a need for an ongoing paradigm shift in how to deal with floods [4]. These difficulties powered the necessity for effective action programmes driven by policy in Europe. According to these limitations, European Commission and the Council of the European Union prompted to put forward the Directive 60/2007/EC, referred to as the Floods Directive [5]. Its purpose is to reduce the adverse effect of floods

to preserve human health, the environment, cultural heritage, economic activity and infrastructure. According to the directive, each member state has to draw up flood risk maps of the river basins and associated coastal areas at risk of flooding and establish flood risk management plans or each catchment [5].

In Portugal, floods are the second most common natural disaster that cause great damage or loss of life. The number of disastrous floods occurred from 1865 to 2010 was 1621 causing 1012 deaths and a partial of 522 victims [6]. In order to manage and mitigate these hazards, Portugal transposed this Directive into their own law (Decree-Law no. 115/22 October 2010) [7]. According to the Directive goals, the Portuguese Agency for the Environment (APA–Agência Portuguesa do Ambiente) identified 23 flood risk zones in many hydrographic basins of mainland Portugal [8]. After the critical flood risk zones were identified, the APA elaborated corresponding cartograms of flood hazard and risk maps [9]. Each map includes areas that may be affected by floods, with a return period equal or greater than 100-year and with a 10-year return period between high probability events [10]. Flood risk maps characterize the different areas by categories, ranging from non-existent risk to very high risk. Several studies were carried out with the aim of flood risk mitigation and flood management [11–15].

In order to attenuate the most significant flood impacts, instead of preventing floods from fully occurring, Reference [16] developed a model based on retention basins, capable of eliminating the areas classified as risky or very risky from those maps. Flood Retention Basins (FRB) are essential in the effective flood management, allowing peak flow attenuation by temporarily storing a certain volume of stream water, and nowadays there exist at least six types of FRB with the purpose to assess flood-control potential beside other possible uses [17–19]. This work comprised the sequential use of engineering formulae and a zoning algorithm embedded in a Geographic Information System, resulting in a number of optimal places to install the detention basins within the critical zones, with a huge diversity of dam wall heights. These results could be influenced by differences in topography, land use, climate, geology or occupation by burnt areas among the sub-catchments [16]. In some critical zones, where the dam wall heights were rather low (<2 m), the attenuation of flood risk could be accomplished through the construction of sustainable flood detention basins [16]. Sustainable flood detention basins are characterized by low construction costs and landscape impact, and frequently can be used to create attractive leisure areas [18,19]. In other critical zones, flood risk attenuation could only be attained if dam walls were taller than 120 meters [16].

Some flood protection solutions have been proposed and the most usual measures of protection are traditionally engineered solutions such as dams, levees and floodwalls. These solutions are essential to safety in many locations, although they can be expensive and can alter flood risk to other locations [20]. A solution proposed instead of a large dam would be to decentralize this into multiple detention basins [17,21], which means small dams could be easily integrated into the natural landscape, with low environmental impact. However, in Reference [17], it was found that this approach may be impracticable, giving rise, instead of a high dam, to several dams of considerable height. Whether in fluvial or pluvial flood mitigation approaches, other proposals to complement the structural solution include green infrastructure investments or Natural Water Retention Measures (NWRM), such as reforestation, installation of grass and riparian buffers, green roofs, porous pavement, urban trees, constructed wetlands, stream restoration, and best-management practices for agriculture and forestry [1,14,22–24]. These types of solutions can contribute to a moderation of flood events by increasing the ability of the landscape to store water or by increasing the ability of channels to convey flood waters. Therefore, it is extremely important on a watershed level to have better forest and wetland management that harness the natural ability of ecosystems to retain water, slowing down and absorbing some of the storm runoff. Forests can also help to reduce flow velocity of flood waters, stabilize banks, land erosion and landslides, and migration of contaminants [25–32].

This study aims to identify "green" protection measures, like reforestation/afforestation or reduction of impermeable/urban areas, complementary to dam construction, that could help reducing the dam wall heights forecasted in the study of Reference [16]. As a way to test the reliability of these protection measures, a statistical model is used to associate the heights of the dams with the

variables of forest cover and impermeable surfaces in order to decrease the height of the dams. The multivariate statistical modeling is based on Partial Least Squares–Path Modeling (PLS-PM). The origin of PLS-PM was in the social sciences [33], but subsequent applications expanded the use of PLS-PM to other areas including the environmental sciences [34–40]. PLS-PM is derived from Structural Equation Models (SEM) which provide additional insights about the dataset's structure, allowing us to comprehend direct as well as indirect interactions between numerous latent (groups of) variables [41]. The PLS-PM assesses the interactions through a combination of Multiple Linear Regression and Principal Component Analysis [42] and represents a substantial improvement of other multivariate statistical models also used in environmental analyses [43–46].

## 2. Materials and Methods

### 2.1. Study Area

The study area is part of the 23 critical flood risk zones identified by the Portuguese Environmental Agency in continental Portugal [47]. A previous investigation in each zone was focused on mitigating the risk of flooding in areas of high and very high risk, using sustainable holding basins according to the 100-year return period (Supplementary Material (Table S1)) [16]. The criterion for evaluating the detention basins was based on Reference [48], which considers as sustainable dam height of ≤8 m. Through this study, results were only obtained for 15 zones and indicated the possibility to install 27 sustainable and 75 non-sustainable detention basins in specific catchments within the critical zones' contributing watersheds. As can be seen in Table 1, of these 15 zones, eight had in their constitution non-sustainable flood detention basins ($h > 8$ m) [16].

**Table 1.** Minimum and maximum sub-basins heights distributed by Critical Zone.

| Critical Zone | Minimum Height of Sub-Basins (m) | Maximum Height of Sub-Basins (m) |
|---|---|---|
| Ponte da Barca | 22.8 | 76.3 |
| Esposende | 0.5 | 116.5 |
| Coimbra | 27.0 | 126.6 |
| Águeda | 0.8 | 57.0 |
| Santarém | 0.5 | 24.6 |
| Tomar | 4.0 | 22.7 |
| Santiago do Cacém | 11.7 | 41.0 |
| Tavira | 3.2 | 36.6 |

It was verified that the most worrying areas are Ponte da Barca, Coimbra and Santiago do Cacém, because these do not present any sub-basins with $h \leq 8$ m.

Therefore, this study focuses on these eight critical zones (Figure 1) and of these, only two are located in international rivers (shared between Portugal and Spain), which are Ponte da Barca, where it integrates with the Lima River, and Santarém, which includes the Tagus River. The remaining six zones are located in national rivers.

These Critical Zones are geographically distributed, in a rather heterogeneous way, throughout continental Portugal and present very different characteristics between them, as depicted in Table 2.

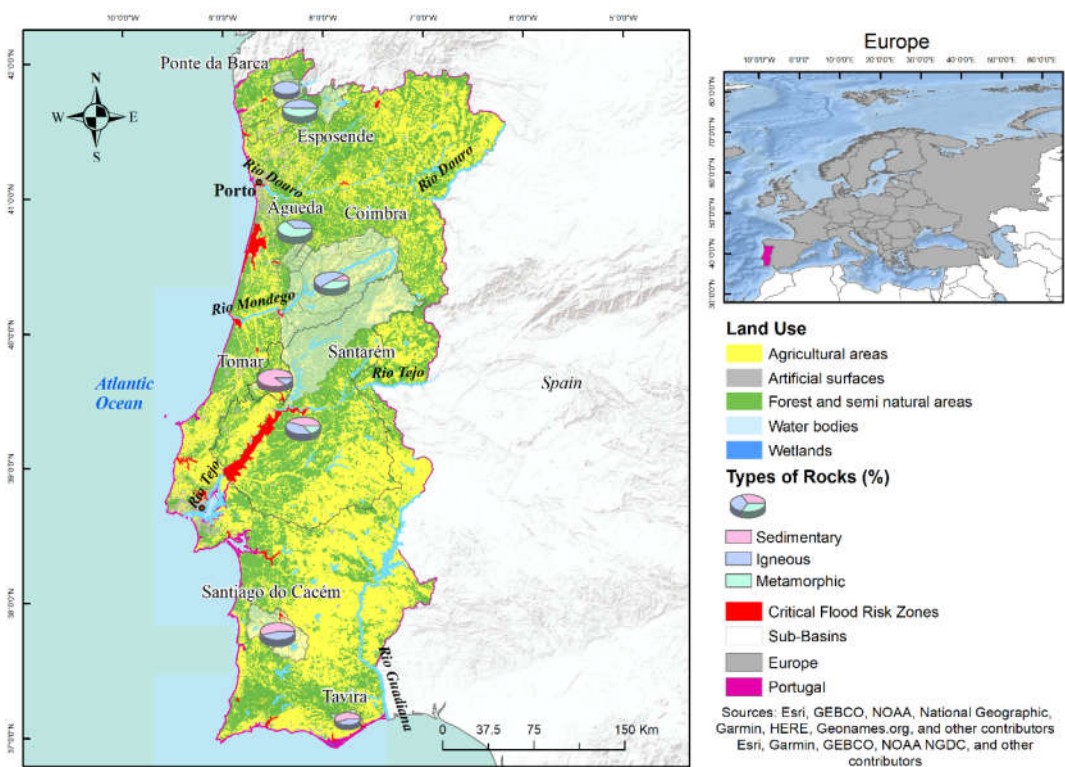

**Figure 1.** Spatial distribution of flood risk critical zones of continental Portugal.

Table 2. Characterization of the eight Critical Zones.

| Critical Zone | Hydrographic Region (HR) | Area (km²) | Maximum Elevation (m) | Slope (%) | Average Temperature (ºC) | Average Annual Rainfall (mm) | Sedimentary Rocks (%) | Igneous Rocks (%) | Metamorphic Rocks (%) | Forest Area (%) | Agricultural Area (%) |
|---|---|---|---|---|---|---|---|---|---|---|---|
| Ponte da Barca | HR1 | 523 | 1401 | 21 | 12 | 2370 | 0 | 99 | 1 | 25 | 31 |
| Esposende | HR2 | 1549 | 1513 | 17 | 13 | 2100 | 0 | 49 | 51 | 28 | 22 |
| Coimbra | HR4 | 4925 | 1993 | 14 | 14 | 1214 | 9 | 52 | 39 | 49 | 29 |
| Águeda | HR4 | 417 | 1043 | 19 | 14 | 1786 | 0 | 36 | 64 | 68 | 22 |
| Santarém | HR5 | 19224 | 1988 | 9 | 15 | 856 | 44 | 41 | 15 | 38 | 42 |
| Tomar | HR5 | 1044 | 394 | 8 | 17 | 1059 | 88 | 12 | 0 | 29 | 42 |
| Santiago do Cacém | HR6 | 1416 | 360 | 3 | 16 | 662 | 54 | 45 | 1 | 24 | 71 |
| Tavira | HR8 | 223 | 507 | 12 | 18 | 730 | 56 | 43 | 1 | 8 | 52 |

According to the classification of Köppen–Geiger, continental Portugal presents a Mediterranean climate. The precipitation is characterized by some frequency of very wet and dry years that affects the hydrological cycle and by consequence the river flows and water resources [49]. The annual average rainfall varies from over 2000 mm, in the northern mountains, to less than 700 mm in the southern plains of Santiago do Cacém. In terms of land use, it is verified that most of the zones have a percentage of forest areas greater than 20% with the exception of Tavira with 8%. Regarding the agricultural areas, the majority presents a percentage greater than forest, except Esposende, Coimbra and Águeda. For decades, there is an increase of population and economic activity intensification in coastal and river areas, especially near the flood plains. Despite the economic value of these areas, they are exposed to frequent floods. During the 1865–2010 period the number of floods that produced negative impacts was 651, while the number of dead, injured or missing people was 546 (>3/year) and the number of evacuated or homeless people was 35,501 (◉250/year) [50]. According to Reference [51], 82% of the hydro-geomorphological events were floods; they were more frequent during the 1936–1967 period and occurred mostly from November to February.

### 2.2. Workflow

The workflow model unfolds in two steps, as can be seen in Figure 2. In Step 1, a range of variables is selected that influences the dam storage. This set of parameters comprises the characteristics of the hydrographical basins, climate indicators, surface runoff, geology and land use. Then, the relationship between these variables is defined using the Partial Least Squares-Path Modeling method (PLS-PM).

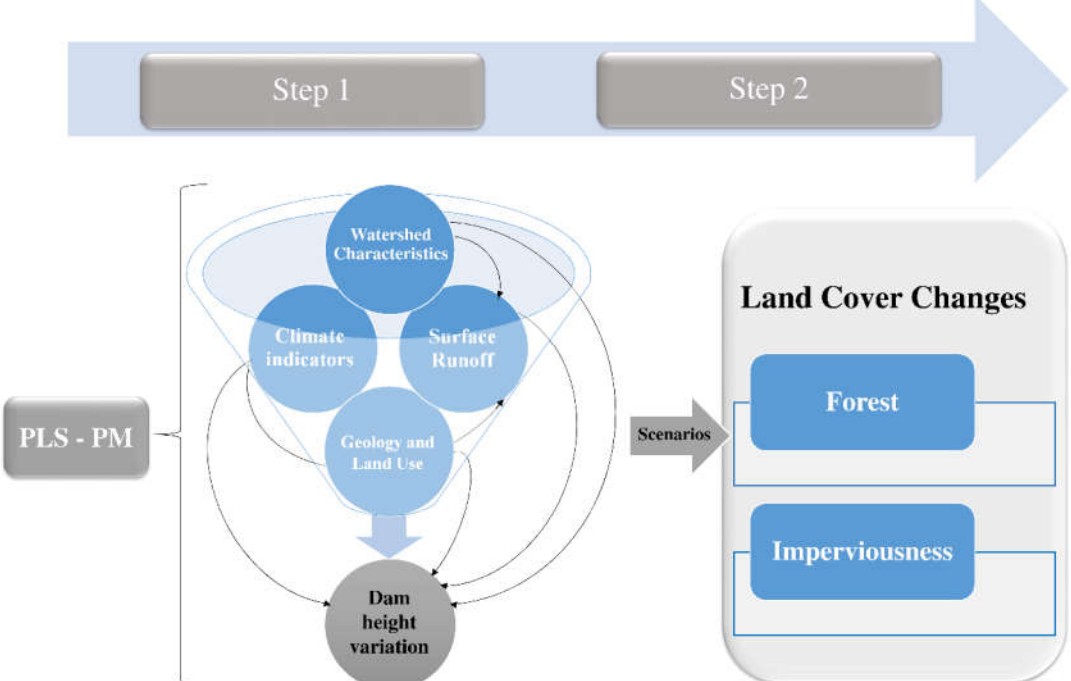

**Figure 2.** Conceptual workflow. The detailed explanation is provided in the text.

The rationale and inventory of variables used in PLS-PM followed the guidelines of an antecedent work [16]. In Step 2, the model equations derived from Step 1 were used to analyse scenarios of dam-height variation in response to anticipated land cover changes mostly related with forest spreading and reduction of impermeable surfaces.

### 2.3. Partial Least Squares-Path Modeling

Partial Least Squares (PLS) models, originally developed by Herman [33,52,53] and Reference [54], are powerful tools for analysing multivariable data [55]. Considered as a soft modeling approach, Partial Least Squares-Path Modeling (PLS-PM) is a statistical model where no strong assumptions, with respect to the distributions, the sample size and the measurement scale, are required [56]. It is used to model causal paths among blocks of variables called latent variables (LV) [34,57]. A PLS-PM consists of two elements (Figure 3), the outer/measurement model, which describes the relationships between the measured variables (MV) and their respective LV (i.e., the loadings and weights), and the inner/ structural model, which describes the relationships between the LV (i.e., the path coefficients) [42,56]. The connections among LV are quantified thorough path coefficients (PC) while the connections between LV and MV are quantified through weights ($w$) [42]. For each regression in the structural model, the amount of variance in the dependent latent variable (also called endogenous) explained by its independent latent variables (exogenous), is indicated by the coefficient of determination ($R^2$). The influence of exogenous on endogenous latent variables is represented by direct or indirect path coefficients, depending on the number of paths linking them [34]. As can be seen in Figure 3, $LV_b$ has just a direct influence on $LV_c$, which is $PC_{b-c}$, while $LV_a$ has direct ($PC_{a-c}$) as well as indirect ($PC_{a-b}$, $PC_{b-c}$) influences. According to [58], the indirect influence is the product of corresponding direct influences (i.e., $PC_{a-b} \times PC_{b-c}$) and the total influence is the sum of direct and indirect influences (i.e., $PC_{a-c} + PC_{a-b} \times PC_{b-c}$).

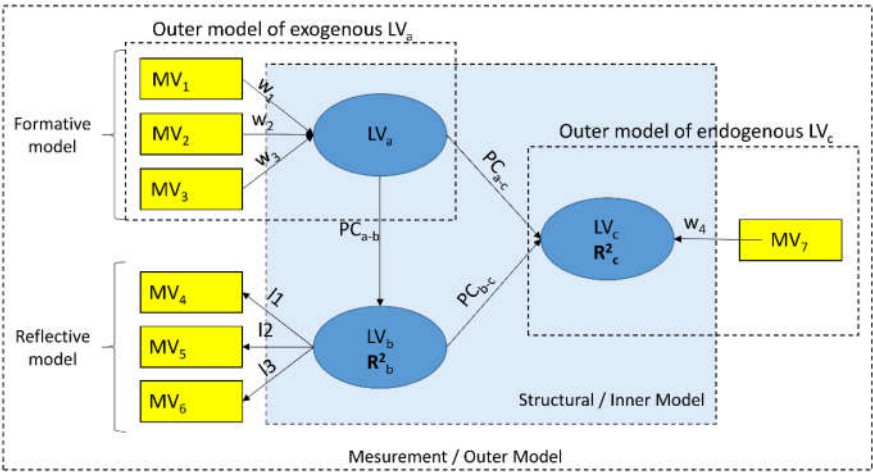

**Figure 3.** Example of a PLS-PM Design. Symbols: MV—measured variables; LV—latent variables; w—weights; l—loadings; PC—path coefficients; R2—coefficient of determination.

The measured (e.g., LVm, Equation (1)) and predicted (e.g., LVp,c; Equation (2)) scores of an LV are calculated as follows:

$$LV_m = \sum_{i=1}^{n}(MV_i \times w_i) \tag{1}$$

$$LV_{p,c} = LV_{m,a} \times PC_{a-c} + LV_{m,b} \times PC_{b-c} \tag{2}$$

The PLS-PM models can be reflective, formative or MIMIC (a mixture of the reflective and formative) [56]. A path model is considered reflective if in the path diagram the arrows go from the LV (factor) to the MV, in other words the LV are considered as the cause of the MF. On the other hand, in a formative model the arrows go from the observed measures to the LV, which means that the MV are considered to be the cause of the LV [36,58,59]. In the present study, PLS-PM was used as MIMIC model, being implemented in the SMART-PLS software [60].

*2.4. Data Preparation*

The PLS-PM input data came from an Excel worksheet comprising $n$ rows and $p$ columns, where $n$ represents the number of sub-basins and $p$ the number of measured variables. Following the results obtained by the work of Reference [16], 75 non-sustainable sub-basins ($n$) were selected. These sub basins were delineated using the ArcHydro [61] software and defined according to the Reference [18] classification, which uses height ($h$) as a key classification variable of flood detention basins, where the structure is sustainable when $h < 8$ m and non-sustainable when $h > 8$ m. A summary of the Measured Variables usage and sources of information is depicted in Table 3. The data for the Excel worksheet was prepared in ArcMap [62] computer package, used in numerous environmental studies (e.g., References [63–80]).

**Table 3.** List of measured variables used as source data for Partial Least Squares–Path Modeling (PLS-PM). Besides identification of variables, their measurement units and description, the table contains indications about usage in the PLS-PM models and on the data sources.

| Measured Variable | Units | Description | Source |
|---|---|---|---|
| Maximum Elevation | m | Maximum elevation obtained from analysis of a Digital Elevation Model (DEM) | http://www.dgterritorio.pt/ |
| Slope | % | Hillside slopes obtained from analysis of a Digital Elevation Model (DEM) | http://www.dgterritorio.pt/ |
| Temperature | °C | Average Annual Temperature | http://www.apambiente.pt |
| Rainfall | mm/year | Total annual precipitation | http://www.apambiente.pt |
| R | | Precipitation Erosivity | http://www.apambiente.pt |
| Drainage Density | km/ km$^2$ | The drainage index explains the complexity and degree of development of a watershed's drainage system. | http://geo.snirh.pt/AtlasAgua/ |
| Surface Flow | m$^3$/s | Annual Average Values | http://geo.snirh.pt/AtlasAgua/ |
| Curve Number (CN) | Dimensionless | Empirical parameter used in hydrology for predicting direct runoff or infiltration from rainfall excess | http://geo.snirh.pt/AtlasAgua/ |
| IMD | Dimensionless | Imperviousness ratio - Relationship between the percentage of change soil sealing and the basin area (more information is provided as Supplementary Material (Table S2)). | https://www.copernicus.eu/ |
| Surface Runoff | mm | Quantity of water in the hydrographic network - precipitation-drainage model according to Temez model. | http://geo.snirh.pt/AtlasAgua/ |
| Sedimentary Rocks | km$^2$/km$^2$ | Percentage of sedimentary rocks in the basin. | http://www.apambiente.pt |
| Igneous Rocks | km$^2$/km$^2$ | Percentage of igneous rocks in the basin. | http://www.apambiente.pt |
| Forest | km$^2$/km$^2$ | Percentage of area covered with forest land uses. | http://www.dgterritorio.pt/ |
| Agricultural | km$^2$/km$^2$ | Percentage of area covered with agriculture land uses. | http://www.dgterritorio.pt/ |
| Shape coefficient ($K_f$) | Dimensionless | Relationship between the mean width of the basin and its axial length. | Equations (2) and (3) and related data |
| Compactness coefficient ($K_c$) | Dimensionless | Relationship between the Perimeter P and the circumference of an equal area circle A, with radius r of the basin. | Equations (2) and (3) and related data |
| Dam Height | m | Calculated dam wall height. | [16] |

For each sub-basin, the mean values of the measured variables were calculated. In the cases that the variables were collected as raster files from the original sources, the calculation of mean values

resorted to the Zonal Statistics as Table (ZST) tool of ArcMap program. The variables land use (agricultural and forest areas) and geology (sedimentary and igneous rocks) were intersected with the sub-basins' shapefile to obtain the coverage of each use and geology in percentage of sub-basin area using the Tabulate Intersection (TI) tool. The percentage of conflict area in each sub-basin was directly incorporated into the Excel worksheet skipping the use of ZST tool. The shape coefficient ($K_f$) and the compactness coefficient ($K_c$) were calculated according to the following equations, respectively:

$$\left\{ \begin{aligned} K_f &= \frac{A}{L^2} \\ K_c &= 0.28 \times \left( \frac{P}{A^{\frac{1}{2}}} \right) \end{aligned} \right. \tag{3}$$

where $A$ (km²) is the sub-basin area, $L$ (km) is the sub-basin length measured along the main water course, and $P$ (km) the sub-basin perimeter.

*2.5. Scenario Analysis*

Flood Risk Management Plans (FRMP) have planned to manage flood risk through prevention, preparation, protection, recovery and learning measures. The first working scenario in this study fits in the protection measures, which fall within the scope of reducing the magnitude of the flood, sometimes by attenuating the flood flow or by reducing the height or flow velocity [22]. Contrarily to structural measures (e.g., construction of dike and dams capable of damping the flood hydrograph), non-structural measures termed green infrastructures (Natural Water Retention Measures–NWRM) are a priority. In this first scenario, the forest area will be increased in order to restore and maintain the aquatic and riparian ecosystems, to promote infiltration and reduce the surface runoff, and expectedly to decrease the height of the dams. Another scenario of land use change is proposed, where the intention is to analyse how the dam height responds when the percentage of agricultural areas is changed. This scenario relates to the public concern about the anthropogenic occupation of floodplains, which has already triggered political decisions for the medium and long terms, involving the relocation of infrastructures, control in the occupation of these areas, and increase of their resilience to floods [22]. According to these alternatives, the second scenario changes land use and occupation in the sub-basins by reducing their imperviousness index (IMD). Therefore, it is intended to verify if the decrease of impermeable zones, which boost runoff, influences the dam height in a significant way.

**3. Results**

*3.1. General Description of Spatial Data*

The spatial distributions of all measured variables are illustrated in the maps of Figures 4–7. The watershed characteristics are illustrated in Figure 4. The CN values (Figure 4a), which predict direct runoff or infiltration from rainfall excess, tend to be large in all sub-basins, and in some zones can be larger than 75. The drainage density (Figure 4b), which measures the complexity and development of the watershed's drainage system, is high in all zones and their sub-basins (CN > 3.5 km/km²), meaning that they are exceptionally well drained. As can be seen in Figure 4c, the slope in the northernmost areas of the country is higher. Coimbra is quite heterogeneous because it comprises sloping areas such as the Serra da Estrela and flat areas such as Santa Comba Dão. Santarém presents a considerable slope due to its proximity to the Serra da Estrela. The slopes are less pronounced to the south of Portugal, except in Tavira where the proximity to the Caldeirão Mountains makes the sub-basins steeper. The maximum altitudes (Figure 4d) are higher in the northern areas of mainland Portugal and decrease to the south, except for the region between Santarém and Coimbra where the Serra da Estrela is located. The $K_c$ values in Figure 4e are higher than 1.5, which means that all sub-basins are moderately flood-prone. The values of $K_f$ can be seen in Figure 4f. Central Portugal is

characterized by lower $K_f$, while basins with smaller shape factors have a lower tendency to originate floods.

The spatial distributions of land use and geology are illustrated in Figure 5. The values of IMD ratio (Figure 5a) are lower in the more urban impermeable areas. As regards the use for agriculture or occupation by forest (Figure 5b,c) the areas are both small in Ponte da Barca and Esposende. In Águeda and Coimbra, the percentage of agricultural area is low but the percentage of forest area is considerable. In Santarém, the forest area is larger than the area used for agriculture, while in Tomar the situation is reversed. Santiago do Cacém and Tavira, in general, present a large percentage of agricultural area and low percentage of forest area. As regards geology (Figure 5d,e), the sedimentary rocks prevail in zones closer to the coast, namely Tomar, Santiago do Cacém and Tavira. The Igneous Rocks are well represented in the centre and south zones of the country.

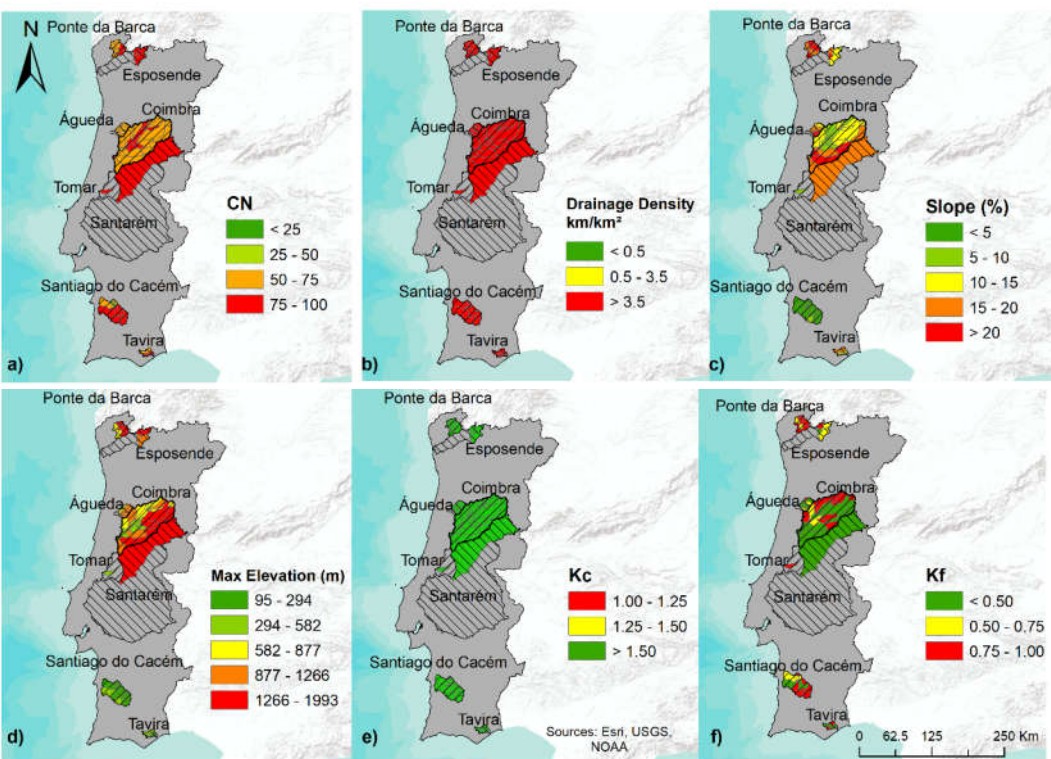

**Figure 4.** Spatial distribution of Watershed Characteristics. Only the variables used in the PLS-PM analyses are represented, namely: (**a**) CN – curve number; (**b**) drainage density; (**c**) terrain slope; (**d**) maximum watershed elevation; (**e**) $K_c$ – compactness coefficient; (**f**) $K_c$ – shape coefficient.

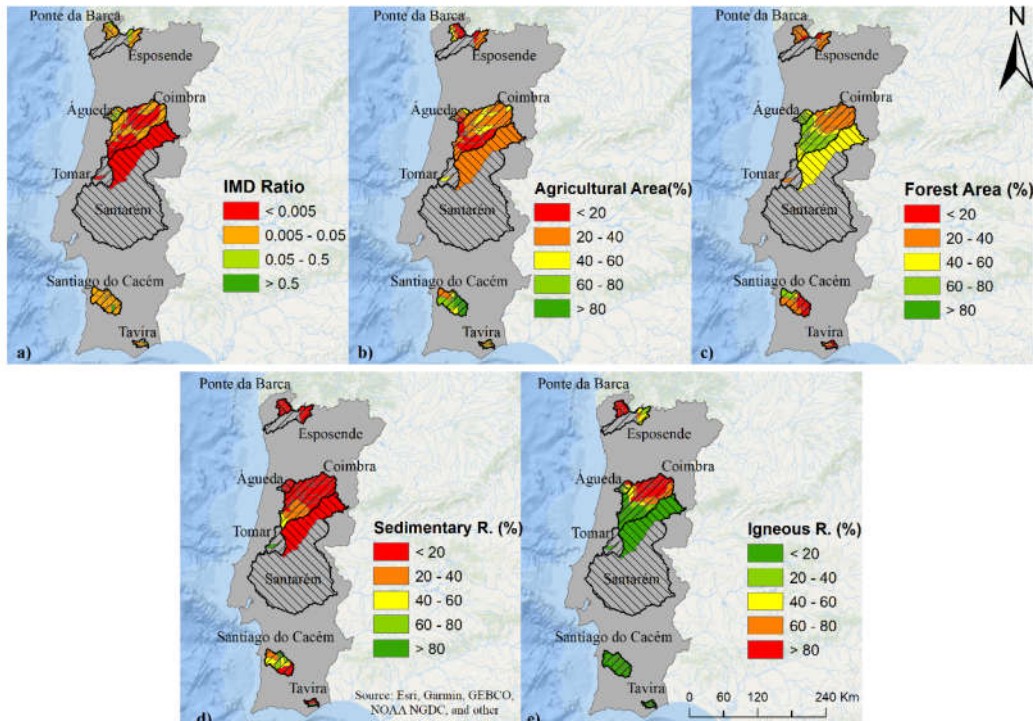

**Figure 5.** Spatial distribution of Land Use and Geology parameters. Only the variables used in the PLS-PM analyses are represented, namely: (**a**) IMD – Imperviousness ratio; (**b**) percentage of agricultural area in the watershed; (**c**) percentage of forest area in the watershed; (**d**) percentage of sedimentary rocks in the watershed; (**e**) percentage of igneous rocks in the watershed.

In Figure 6 the climate indicators can be visualized. Regarding rainfall erosion (Figure 6a), the most affected areas are Ponte da Barca, Esposende, Águeda and Tavira. With regard to total annual precipitation (Figure 6b), the trend is to decrease from north to south of the country. The temperature (Figure 6c) increases from north to south, except for some sub-basins of Coimbra and Tomar that present values similar to the zones of the south.

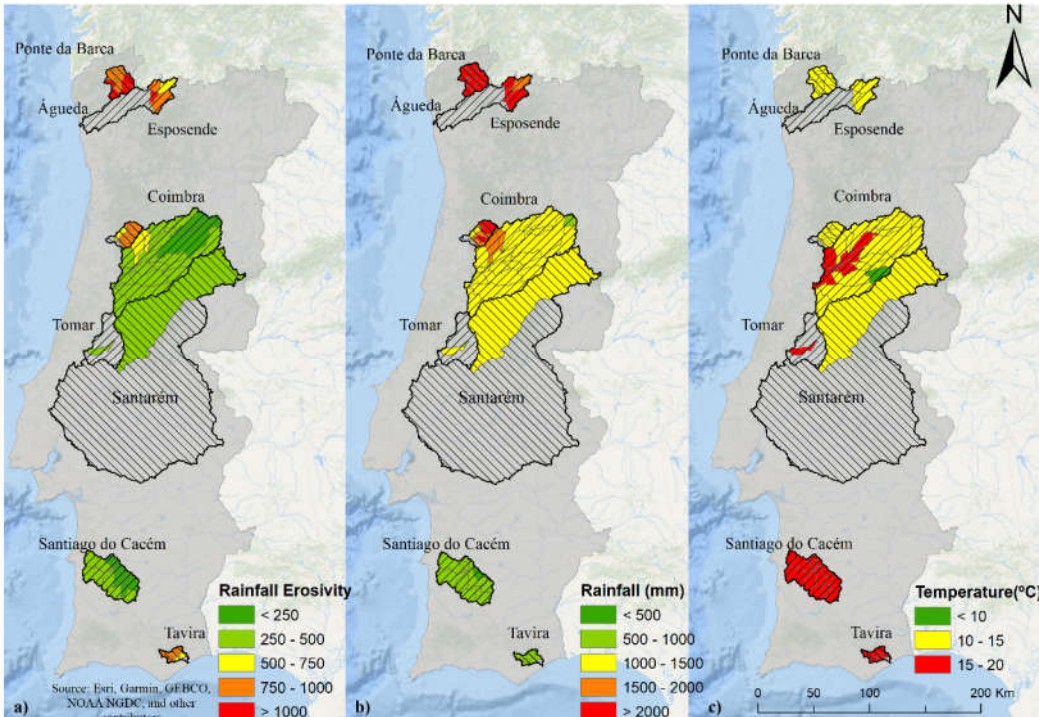

**Figure 6.** Spatial distribution of climate indicators. Only the variables used in the PLS-PM analyses are represented, namely: (**a**) rainfall erosivity; (**b**) annual rainfall; (**c**) mean annual temperature.

Runoff (Figure 7a,b) follows the downward trend from north to south. The spatial distribution of the dam heights is illustrated in Figure 8. The figure shows that the areas with the greatest heights are Esposende and Coimbra, where dam heights can exceed 80 m. Ponte da Barca also presents some sub-basins with heights between the 60 and 80 m. In Águeda and Santiago do Cacém the heights do not exceed 60 m while in the remaining zones the heights are ≤40 m.

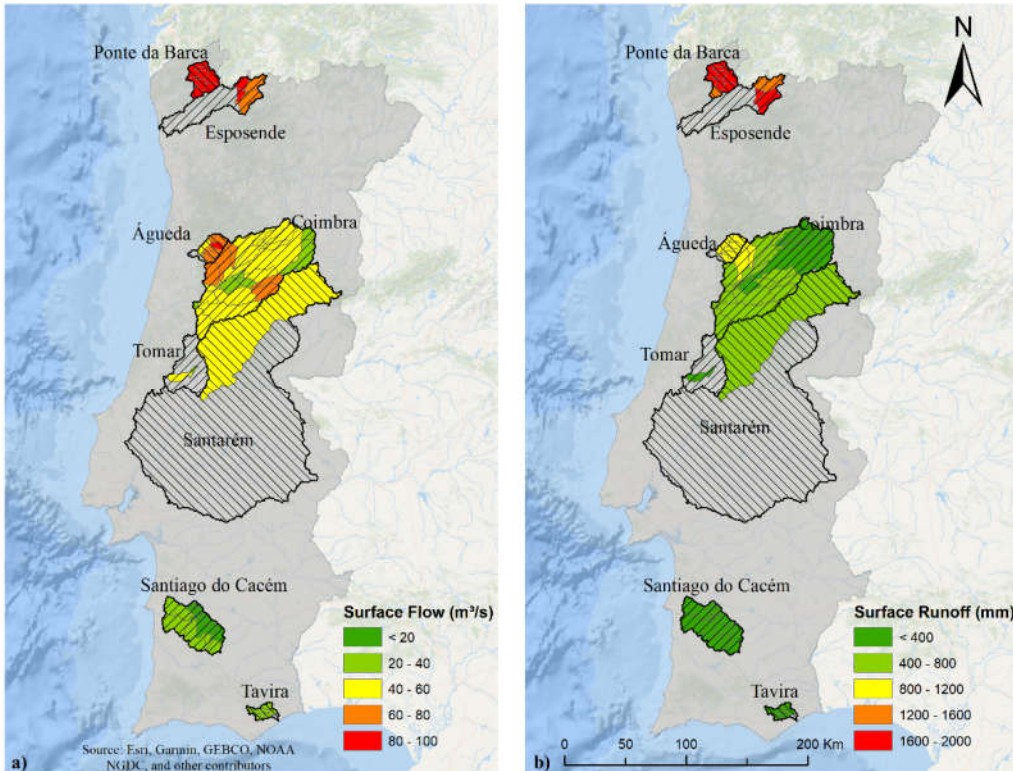

**Figure 7.** Spatial distribution of Surface Runoff. Only the variables used in the PLS-PM analyses are represented, namely: (**a**) Surface flow expressed as total surface discharge (m³/s); (**b**) surface runoff expressed as surface discharge normalized by catchment area.

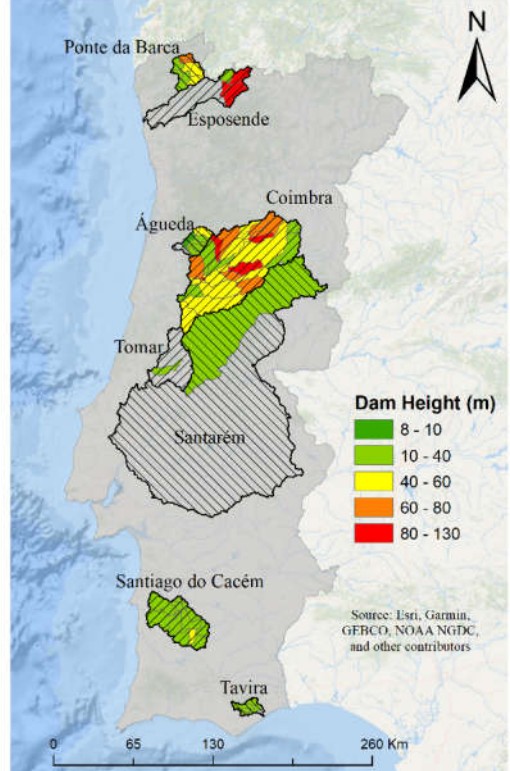

**Figure 8.** Spatial distribution of dam wall height.

### 3.2. Results of Partial Least Squares-Path Modelling

The PLS-PM model for the eight Critical Zones, more specifically the 75 sub-basins with dam height >8 m is provided as Supplementary Material (Table S3) and was compiled from the previous work by Reference [16]. This model relates the MVs to their LVs through the MIMIC approach, i.e., through a mixture of reflective and formative models [56]. The reflective model requires an assessment of reliability and validity to support the inclusion of construct measures in the path model [57]. Reliability was measured by the Dillon–Goldstein Rho, which must be larger than 0.7 [56]. Validity was confirmed by the Average Variance Extracted (AVE), which must be larger than 0.5 [57]. The formative model requires an assessment of multi collinearity among measured variables through analysis of variance inflation factors (VIF), which must be lower than 5 for predictive purposes [58].

The reliability, validity and VIF constraints were confirmed in this study. The PLS-PM model is portrayed in Figure 9. The variances explained by the model are high for the endogenous LV "Surface Runoff" ($R^2 \approx 0.9$) but relatively low for the LV "Dam Height" ($R^2 \approx 0.3$). Despite the low score of LV "Dam Height" the model is robust and reliable because the sample is large. All the path coefficients expose positive causal effects with the LV "Dam Height", except the LV "Climate Indicators" which are negative. The weights of most MVs are positive, which means that the LVs increase for increasing values of their formation variables. The negative weights mean that the increase of correspondent MVs contribute to the decrease of dam height.

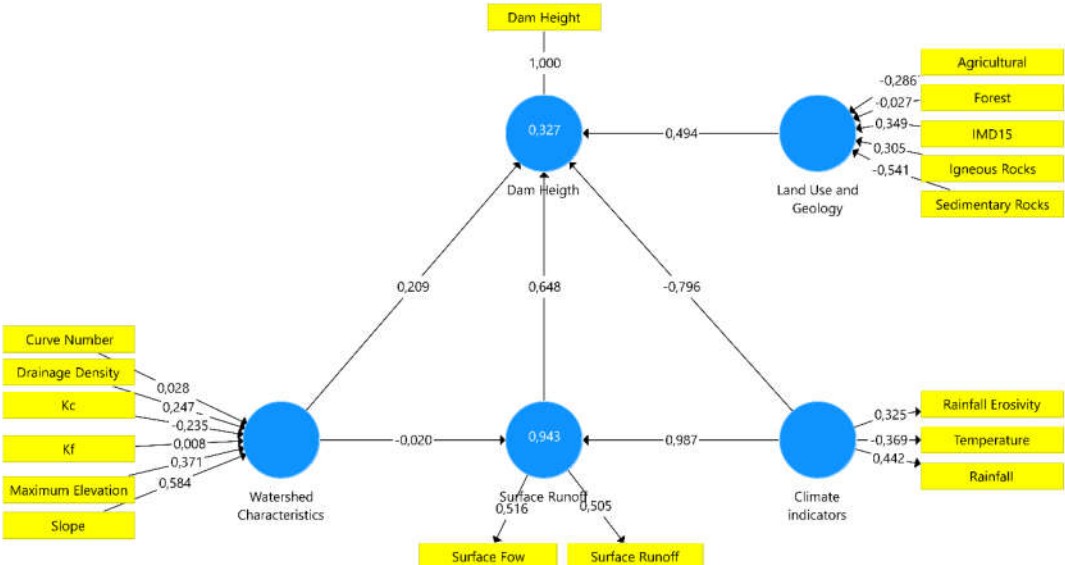

**Figure 9.** PLS-Path Model used in the present study to set up causal effects between watershed characteristics, surface runoff, climate indicators, land use and geology and dam wall height.

The imperviousness index ($w$ = 0.349) and sedimentary rocks ($w$ = −0.541) are the MVs contributing stronger to the formation of LV "Land use and Geology", due to their larger weights. Slope ($w$ = 0.584), Surface Flow or Runoff ($w$ = 0.516 and $w$ = 0.505) and Rainfall ($w$ = 0.442) are other measured variables that contribute strongly to the formation of their respective LVs. The Equations (4)–(8) describe the outer model for "Watershed Characteristics", "Land Use and Geology", "Runoff", "Climate Indicators" and "Dam Height", respectively.

$$\begin{aligned} Watershed\ Characteristics \\ = CN \times (0.028) + Drainage\ Density \times (0.247) + K_c \times (-0.235) \\ + K_f \times (0.008) + Max.Elevation \times (0.371) + Slope \times (0.584) \end{aligned} \quad (4)$$

$$Land\ Use\ and\ Geology$$
$$= Agricultural \times (-0.286) + Forest \times (-0.027) + IMD \times (0.349) \quad\quad (5)$$
$$+ Igneous\ R. \times (0.305) + Sedimentary\ R. \times (-0.541)$$

$$Runoff = Surface\ Flow \times (0.516) + Surface\ Runoff \times (0.505) \quad\quad (6)$$

$$Climate\ Indicators$$
$$= Rainfall\ Erosivity \times (0.325) + Temperature \times (-0.369) \quad\quad (7)$$
$$+ Rainfall \times (0.442)$$

$$Dam\ Height\ Variation = Dam\ Height \times (1.000) \quad\quad (8)$$

Equations (9) and (10) describe the inner model for "Runoff" and "Dam Height".

$$Runoff = Watershed\ Characteristics \times (-0.020) + Climate\ Indicators \times (0.987) \quad\quad (9)$$

$$Dam\ Height\ Variation$$
$$= Watershed\ Characteristics \times (0.209)$$
$$+ Land\ Use\ and\ Geology \times (0.494) + Runoff \times (0.648) \quad\quad (10)$$
$$+ Climate\ Indicators \times (-0.796)$$

### 3.3. Forecasted Dam Heights–Scenarios

Among the protection measures outlined in the Portuguese Hydrographic Region Management Plans (https://www.apambiente.pt/), actions are present that rely on structural interventions (for example, construction of dams with damping capacity of the flood hydrograph), but other solutions are also indicated that are based on green measures (Natural Water Retention Measures–NWRM). Therefore, in order to explore the possibility to reduce the dam wall heights, three land-use-change scenarios were tested that predicted forest + agriculture area increase and imperviousness (IMD ratio) decrease. The scenarios were formulated according to the possibility to retain a part of runoff, increasing the retention capacity of catchments by enlarging the forested and agricultural areas or reducing imperviousness in the urban areas. The detailed scenario analysis is presented in Table 4. The results show that in response to increasing forest and agricultural areas, the heights of the dams decrease, but not substantially because the heights remained very high, hampering the possibility of using green measures as a complement to the structural measures. The same happens with imperviousness. When it decreases the heights also decrease but residually. On average, the heights decrease 0.37% in the forest first scenario, 1.4% in the forest second scenario, 0.3% in the IMD third scenario, 0.9% in the fourth IMD scenario, and finally the heights decrease ◎7% in the fifth scenario. This last scenario is only hypothetical and theoretical because it may not be possible to make it real.

## 4. Discussion

The Floods Directive (Directive 60/2007/EC) encouraged the EU member states to evaluate areas at risk of flooding, to map the flood extent, assets and human lives at risk in these areas, and to take adequate and coordinated measures to reduce flood risk. Some reports assure that climate change may result in sea-level rises, which are expected to induce more extreme weather events and increased flood risks as a consequence [2,3]. Therefore, it is important for EU member states to take into consideration climate change as well as sustainable land use practices in flood risk management [81]. Bearing in mind this concern, the prime objective of this research was to develop a statistical model that establishes a relationship between biophysical parameters of catchments and dam wall height. This relationship aimed to find a route to reduce the dam wall height through changes in potentially modifiable parameters such as land use in rural areas and imperviousness in urban centers. A successful relationship would allow controlling floods with sustainable (small height) detention basins.

**Table 4.** Results of scenario analysis in the 75 sub-basins of the eight critical flood risk zones.

| Critical Flood Risk Zones | No of Sub-basins | Average Height Predicted by the Model (m) | | 1st Scenario (+30% of Forest Areas) | | 2nd Scenario (+100% of Forest Areas) | | 3rd Scenario (−30% of Impermeable Areas) | | 4th Scenario (−100% of Impermeable Areas) | | 5th Scenario (−50% of Slope) | |
|---|---|---|---|---|---|---|---|---|---|---|---|---|---|
| | | Minimum Height (m) | Maximum Height (m) | Minimum Height (m) | Maximum Height (m) | Minimum Height (m) | Maximum Height (m) | Minimum Height (m) | Maximum Height (m) | Minimum height (m) | Maximum Height (m) | Minimum Height (m) | Maximum Height (m) |
| Águeda | 9 | 39.77 | 60.93 | 39.51 | 60.56 | 38.91 | 59.71 | 39.76 | 60.79 | 39.74 | 60.48 | 37.09 | 55.01 |
| Coimbra | 30 | 31.09 | 114.61 | 31.02 | 114.26 | 30.85 | 113.45 | 31.09 | 100.43 | 31.08 | 67.35 | 28.74 | 110.8 |
| Esposende | 4 | 44.33 | 61.44 | 44.16 | 61.31 | 43.76 | 61.00 | 44.32 | 61.39 | 44.30 | 61.26 | 40.94 | 56.49 |
| Ponte da Barca | 10 | 46.56 | 68.38 | 46.54 | 68.27 | 46.50 | 68.00 | 46.48 | 68.12 | 46.31 | 67.53 | 43.52 | 61.62 |
| Santarém | 1 | 48.44 | 48.44 | 48.21 | 48.21 | 47.68 | 47.68 | 48.44 | 48.44 | 48.44 | 48.44 | 44.49 | 44.49 |
| Santiago do Cacém | 13 | 11.51 | 36.33 | 11.48 | 36.30 | 11.42 | 36.21 | 11.36 | 35.57 | 11.02 | 34.73 | 11.06 | 35.42 |
| Tavira | 6 | 15.61 | 43.32 | 15.57 | 43.32 | 15.47 | 43.31 | 15.59 | 43.14 | 15.54 | 42.70 | 13.21 | 40.04 |
| Tomar | 2 | 21.32 | 22.30 | 21.21 | 22.28 | 20.96 | 22.24 | 21.32 | 22.25 | 21.32 | 22.12 | 19.22 | 20.16 |

Examples of runoff or flood peak reductions in response to forestation are numerous in recent scientific literature. For example, vegetation cover in the upper Du watershed in China was significantly improved after the implementation of the Grain-for-Green project [82]. An analysis of variance indicated that reforestation resulted in a significant reduction in runoff and sediment transport. Another study in the Chao Phraya River Basin and based on numerical models [83] showed that specific non-structural measures (reforestation, wetlands, and the combination of both) had considerable potential to reduce peak discharges and flood volumes. Indeed, it was suggested that integration of these proposed non-structural flood countermeasures with the existing countermeasures in the Chao Phraya River Basin would be the most practical way to cope with the challenges of future flood disasters. Bearing in mind these results, an expectable outcome from our modeling exercise would be a substantial reduction of dam wall heights in response to changes in land use of catchments. However, that did not occur. In most critical zones of continental Portugal, the flood control based on structural measures relied on construction of medium to large structures, namely dams with 9 to 127 m high walls. In order to reduce these values and convert these engineering structures into sustainable detention basins, all variables that influence flood volumes were studied through a PLS-PM statistical model. The results have shown that Slope ($w > 0.5$), Rainfall ($w > 0.4$) and Sedimentary Rocks ($w > 0.5$) are the most weighted measured variables in the model. These results mean that these measured variables are the ones contributing most to the dam wall height variation, but these variables cannot be easily changed, at least by human intervention. On the other hand, the measured variables Curve Number, $K_f$ and Forest seem to play a less prominent role in the model ($w < 0.1$), in spite of their probable modification by man. In order to achieve our objective, the first scenarios added 30% to forest areas and reduced 30% of impermeable zones, but the PLS-PM model indicated that these measured variables barely interfere in the decrease of the heights. In a second round, unrealistic values such as doubling the forest areas or eliminating the impermeable surfaces were tested, but reduction of the heights was not enough to bring them to sustainable values according to Reference [48]. Finally, when slope, rainfall or flow have changed, which are variables that humans can barely interfere, the heights have dropped moderately (e.g., ≈7% for 50% of slope decrease).

Some studies have already suggested that land cover is apparently ineffective at regulating floods larger than the 10-year flood [84]. In a study in North Carolina spanning the 1930–2000 period, Lecce and Kotecki [85] found no relation between human-induced land cover changes and flood severity in their analysis of relations among river flow, population growth, number of housing units, and area under cultivation. On the other hand, catastrophic floods in China have led to investments in costly reforestation projects, with little evidence of their effectiveness in reducing floods [86]. Large floods seem to be determined by other large-scale drivers such as precipitation and temperature [87], being inherently linked to the return period concept in the case of precipitation. Various authors suggest that natural and anthropogenic features can alter flood characteristics, but these influences decrease as flood-return period increases [88,89]. For example, while working in a small watershed in Georgia, Magilligan and Stamp [90] reconstructed past land cover and modelled hydrologic alterations derived therefrom, recognizing greater temporal variability among two-year floods than among 100-year floods. Two studies carried in forested watersheds [91,92] found no evidence of reduced peak runoff volumes for the 100-year flood. Therefore, large floods may not be managed effectively by manipulating landscape structure, for example through reforestation [84]. For large floods, the solution may inevitably require the installation of flood detention basins and may even need to disperse the big flood into many small floods through simultaneous control of stream flow at various river sections within the watershed [19,93,94]. Eventually, the conjunctive management of floods using structural and non-structural measures is the best route to follow in most cases, as proposed in this study.

## 5. Conclusions

A previous study was carried out on the 23 critical flood risk zones of continental Portugal, where structural measures, namely detention basins, were proposed for installation in 15 of them. In

eight of these zones the dams were rather high (>8 m); all variables that could influence the dam height were assembled and studied in the present research. A PLS-PM model was developed, whereby a relationship was established between these measured variables and the dam-wall heights. The aim was to verify if changes in specific catchment variables, such as forest occupation or imperviousness of urban areas, would result in lower heights. With a 30% increase in forested areas, the heights fell, but insignificantly. On average, the heights dropped about 0.2 m. The same happened when impermeable zones were reduced by 30%. In this case the heights decreased 0.2 m on average. Even when the tested scenarios were unrealistic (for example, double the forest areas), the heights did not fall significantly and remained very high. Finally, even when the predictions were based on variables that can barely be changed through human intervention (for example, reducing average catchment slope) the heights did not go down to values of sustainable detention basins (e.g., ≤8 m). These results open a discussion on whether large floods can be effectively managed by manipulating landscape structure, for example through reforestation.

**Supplementary Materials:** The following are available online at www.mdpi.com/xxx/s1, Table S1 – dam wall heights required to minimize high and very high flood risks in critical zones of continental Portugal, estimated by [16], Table S2: Data on Imperviousness ratio (relationship between the percentage of change soil sealing and the basin area), Table S3: PLS-PM data.

**Author Contributions:** Conceptualization, D.P.S.T. and F.A.L.P.; Methodology, D.P.S.T. and F.A.L.P.; Validation, F.A.L.P.; Supervision, F.A.L.P. and L.F.S.F.; Project Administration, R.M.V.C.; Funding Acquisition L.F.S.F., F.A.L.P. and R.M.V.C.; Software implementation, D.P.S.T. and J.P.M., Data Curation and Processing, D.P.S.T.; Writing-Original Draft Preparation, D.P.S.T.; writing—review and editing, F.A.L.P.

**Funding:** This research was funded by the INTERACT project: "Integrated Research in Environment, Agro-Chain and Technology", no. NORTE-01-0145-FEDER-000017, in its line of research entitled BEST – "Bioeconomy and Sustainability", co-financed by the European Regional Development Fund (ERDF) through NORTE 2020 (North Regional Operational Program 2014/2020). For authors integrated in the CITAB research centre, it was further financed by the FEDER/COMPETE/POCI– Operational Competitiveness and Internationalization Programme, under Project POCI-01-0145-FEDER-006958, and by National Funds of FCT–Portuguese Foundation for Science and Technology, under the project UID/AGR/04033/2019. For the author integrated in the CQVR, the research was additionally supported by National Funds of FCT–Portuguese Foundation for Science and Technology, under the project UID/QUI/00616/2019.

**Conflicts of Interest:** The authors declare no conflicts of interest. The funders had no role in the design of the study; in the collection, analyses, or interpretation of data; in the writing of the manuscript or in the decision to publish the results.

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
