# Peer review of "Can Land Cover Changes Mitigate Large Floods? A Reflection Based on Partial Least Squares-Path Modeling"

_water, doi:10.3390/w11040684_

Round 1
Reviewer 1 Report
Authors of this manuscript have created an interesting model analysis of possible solutions to fluvial flood management in Portugal. This approach is pertinent and interesting, and can be applied elsewhere in the world. Although the manuscript has a clear and state of the art approach, the manuscript includes several grammatical errors throughout that make it difficult to follow. As it stands this script needs rewording before it is suitable for publication. Please ensure that a native English speaker is able to read over the work, to ensure if flows in academic English
The authors need to differentiate in the introduction between pluvial and fluvial flood mitigation approaches. Although they clearly interlink, some of the measures discussed (line 85-88), are more associated with direct source control runoff reduction, which is broadly different to that of fluvial, catchment runoff management.
The methodology is logical. I particularly like the conceptual workflow. On line 115, you state that h<8m is considered sustainable. Please make it clear that you are referring to a dam height around the sub-basin. The authors suggest that they are going to analyse the role of different NWRM (by introducing this idea in the introduction), however Step 2 focuses on just forest and imperviousness. This is a rather simplification of NWRM. If you are going to analyse the role of forest and imperviousness (as is the case, based on your results section), I recommend changing the focus in the introduction away from NWRM/BMP approaches, to solely focus on the impact of forest cover and impermeable surfaces/urbanization. It would also be worthwhile knowing the resolution of the provided datasets (particularly imperviousness ratio), as this will have a considerable weighting on the final results.
I would like to see more constructive analysis of the data. The discussion section is a little light. You outline that aggregating large flow across multiple simultaneous control systems is efficient – can this be incorporated into the model? Although some of these measures are referred to in the introduction, they are not incorporated into the model.
I have made a number of suggestions below for your consideration.
Specific Comments
Line 11 - Remove the before Natural Water Retention Measures
Line 12 – Remove the before detention basins.
Line 21- Change the wording from “man” to “human activity”
Reword and break up Paragraph 2 (Line 44-Line 93)
Line 94 – You state that the study aims to find the “green” protection measures that are complementary. Are these those that are referred to on line 86, outlined as NWRM? If so – please state this. If not, please provide an outline of the measures to be studied.
Line 97 – You have already provided an acronym for NWRM. It does not need to be written out again in full.
Line 110 – There appears to be a change in spacing between paragraphs? If this is the case, please change accordingly, in line with the journal requirements.
Line 113 – Is there a specific storm duration for the 1 in 100 year return period, or is it the critical storm duration?
Line 115 – There appears to be a format error with h < 8m. Please ignore if this is simply a display error from my end.
Line 358 – Change stablish to establish
Author Response
Reviewer #1:
Please see the yellow shaded lines in the revised manuscript, which describe the masjor changes to the original manuscript, requested by the reviewer. Besides these changes, the entire manuscript was thoroughly revised for language and grammar. Because the changes were many we decided not to highlight them
We very much thank the comments and suggestions that were all welcome and thoroughly addressed. We did our utmost to comply with all requests with the purpose to improve the revised version.
Authors of this manuscript have created an interesting model analysis of possible solutions to fluvial flood management in Portugal. This approach is pertinent and interesting, and can be applied elsewhere in the world. Although the manuscript has a clear and state of the art approach, the manuscript includes several grammatical errors throughout that make it difficult to follow. As it stands this script needs rewording before it is suitable for publication. Please ensure that a native English speaker is able to read over the work, to ensure if flows in academic English
The manuscript was substantially improved for language and grammar, by a professional reviewer.
The authors need to differentiate in the introduction between pluvial and fluvial flood mitigation approaches. Although they clearly interlink, some of the measures discussed (line 85-88), are more associated with direct source control runoff reduction, which is broadly different to that of fluvial, catchment runoff management.
The sentence has been reformulated.
The methodology is logical. I particularly like the conceptual workflow. On line 115, you state that h<8m is considered sustainable. Please make it clear that you are referring to a dam height around the sub-basin.
The sentence has been reformulated.
The authors suggest that they are going to analyse the role of different NWRM (by introducing this idea in the introduction), however Step 2 focuses on just forest and imperviousness. This is a rather simplification of NWRM. If you are going to analyse the role of forest and imperviousness (as is the case, based on your results section), I recommend changing the focus in the introduction away from NWRM/BMP approaches, to solely focus on the impact of forest cover and impermeable surfaces/urbanization.
We took into account your suggestion and changed the focus solely to the directed variables on study.
It would also be worthwhile knowing the resolution of the provided datasets (particularly imperviousness ratio), as this will have a considerable weighting on the final results.
It has been added to the supplementary material the sheet of imperviousness ratio calculation and cited at Table 3.
I would like to see more constructive analysis of the data. The discussion section is a little light.
The discussion section was expanded to include more constructive analysis of the modeling results and more supporting references.
You outline that aggregating large flow across multiple simultaneous control systems is efficient – can this be incorporated into the model? Although some of these measures are referred to in the introduction, they are not incorporated into the model.
Regarding to this measure, the text on lines 75 - 78 was amended because the idea that was intended to be transmitted was not clear. This measure was not used firstly, because it goes out of the scope of thi study and secondly because this is a measure already studied by [17] where it was verified that there would not be a great decrease of the Heights of the dams through the decentralization of one big dam into multiple detention basins.
I have made a number of suggestions below for your consideration.
1- Line 11 - Remove the before Natural Water Retention Measures
“The” has been removed.
2- Line 12 – Remove the before detention basins.
“The” has been removed.
3- Line 21- Change the wording from “man” to “human activity”
The change was made, as suggested.
4- Reword and break up Paragraph 2 (Line 44-Line 93)
The paragraph was broken and the text changed.
5- Line 94 – You state that the study aims to find the “green” protection measures that are complementary. Are these those that are referred to on line 86, outlined as NWRM? If so – please state this. If not, please provide an outline of the measures to be studied.
The specific protection measures has been referred at the text.
6- Line 97 – You have already provided an acronym for NWRM. It does not need to be written out again in full.
This sentence has been replaced and NWRM is no longer included in it.
7- Line 110 –There appears to be a change in spacing between paragraphs? If this is the case, please change accordingly, in line with the journal requirements.
It has been corrected.
8- Line 113 – Is there a specific storm duration for the 1 in 100 year return period, or is it the critical storm duration?
The 100 year return period refers the probability of occurring being 1 in 100, or 1% in any one year.
9- Line 115 – There appears to be a format error with h < 8m. Please ignore if this is simply a display error from my end.
The sentence has been reformulated, since the previous sentence could raise doubts.
10- Line 358 – Change stablish to establish
Changed stablish for establish.
Reviewer 2 Report
1. General Comments
In general, the topic is interesting. I would like to congratulate the authors for a considerable amount of work that they have done. Indeed, floods are a severe threat for many countries and therefore looking for sustainable mitigation measures is in high interest.
However, the manuscript needs some further improved before to be accepted for publication. In general, there are still some occasional grammar errors through the manuscript especially the article ‘’the’’, ‘’a’’ and ‘’an’’ is missing in many places, please make a spellchecking in addition to these minor issues. The reviewer has listed some specific comments that might be helpful of the authors to further enhance the quality of the manuscript. Please consider the specific comments listed below!
Specific Comments
2.1. Abstract
The abstract is well written. The structure is fine.
I would suggest using only dam heights instead of dam wall heights.
2.2. Introduction
· This section is well written; the objectives are explicitly stated.
2.3. Methods
· This section is also well written.
· Line 115, there is an overlapping text.
· Please explain what was the data (hydrological data) resolution used in this study (hourly, daily…) and why?
· Figure 1, artificial and agriculture surfaces have almost the same colour; please change the colour to make the map readable.
· Perhaps you need to increase the font size in figure 1; it is a little bit difficult to read.
· Table 2, temepertures refere to mean values?
· Table 2, you have presented the type of rocks in different zones, but the spatial location is not shown on the map!
2.4. Results
· This section is well written, and the results are adequately explained.
· Please increase the font size in the following figures: 4, 5, 6, 7, and 8.
2.5. Discussion
· This section is well written; I don’t have any comments.
2.6. Conclusion
· This section is well written.
2.7. References
Please check the references in the text and the list; some of them are not according to the journal style.
The author's contribution description is missing; please write it; it is mandatory for this journal.
Author Response
Reviewer #2:
Please see the yellow shaded lines in the revised manuscript, which describe the masjor changes to the original manuscript, requested by the reviewer. Besides these changes, the entire manuscript was thoroughly revised for language and grammar. Because the changes were many we decided not to highlight them
We very much thank the comments and suggestions that were all welcome and thoroughly addressed. We did our utmost to comply with all requests with the purpose to improve the revised version.
General Comments
In general, the topic is interesting. I would like to congratulate the authors for a considerable amount of work that they have done. Indeed, floods are a severe threat for many countries and therefore looking for sustainable mitigation measures is in high interest.
However, the manuscript needs some further improved before to be accepted for publication. In general, there are still some occasional grammar errors through the manuscript especially the article ‘’the’’, ‘’a’’ and ‘’an’’ is missing in many places, please make a spellchecking in addition to these minor issues. The reviewer has listed some specific comments that might be helpful of the authors to further enhance the quality of the manuscript. Please consider the specific comments listed below!
1- Abstract - The abstract is well written. The structure is fine. I would suggest using only dam heights instead of dam wall heights.
Changed dam wall heights to dam heights in lines 18 and 25.
2- Introduction - This section is well written; the objectives are explicitly stated.
3- Methods - This section is also well written.
3.1- Line 115, there is an overlapping text.
Now in line 111, I think it has been adjusted.
3.2- Please explain what was the data (hydrological data) resolution used in this study (hourly, daily…) and why?
The modeling of flood control based on detention basins used daily discharge rates that captured the flood events. These values were then incorporated in the PLS-pm model as to seek for their influence on dam height.
3.3- Figure 1, artificial and agriculture surfaces have almost the same colour; please change the colour to make the map readable.
It has been changed.
3.4- Perhaps you need to increase the font size in figure 1; it is a little bit difficult to read.
The font size has been increased.
3.5- Table 2, temepertures refere to mean values?
Yes, it was already changed in the Table 2.
3.6- Table 2, you have presented the type of rocks in different zones, but the spatial location is not shown on the map!
Information about the Type of rocks has been added to the figure 1.
4- Results - This section is well written, and the results are adequately explained.
4.1- Please increase the font size in the following figures: 4, 5, 6, 7, and 8.
The font size of the figures 4, 5, 6, 7 and 8 has been increased.
5- References - Please check the references in the text and the list; some of them are not according to the journal style.
Some References in the text and list has been changed according to the journal style.
6- The author's contribution description is missing; please write it; it is mandatory for this journal.
Contribution description added to the paper at line 414.